# Performance Optimization of High Specific Speed Centrifugal Pump Based on Orthogonal Experiment Design Method

**Zikang Li [1], Hongchang Ding [1,\*], Xiao Shen [1] and Yongming Jiang [2]**

[1]   College of Mechanical and Electronic Engineering, Shandong University of Science and Technology, Qingdao 266590, China; powerlzk@163.com (Z.L.); shenxiao555@163.com (X.S.)

[2]   Group of Shanghai Liancheng, Shanghai 201812, China; m13256922556@163.com

\*   Correspondence: dhchang@sdust.edu.cn; Tel.: +86-13853292764

**Abstract:** A high specific speed centrifugal pump is used in the situation of large flow and low head. Centrifugal pump parameters need to be optimized in order to raise its head and efficiency under off-design conditions. In this study, the orthogonal experiment design method is adopted to optimize the performance of centrifugal pump basing on three parameters, namely, blade outlet width $b_2$, blade outlet angle $\beta_2$ and blade wrap angle $\varphi$. First, the three-dimensional model of the centrifugal pump is established by CFturbo and SolidWorks. Then nine different schemes are designed by using orthogonal table, and numerical simulation is carried out in CFX15.0. The final optimized combination of parameters is $b_2 = 24$ mm, $\beta_2 = 24°$, $\varphi = 112°$. Under the design condition, the head and efficiency of the optimized centrifugal pump are appropriately improved, the increments of which are 0.74 m and 0.48%, respectively. However, the efficiency considerably increases at high flow rates, with an increase of 6.9% at 1.5 $Q_d$. The anti-cavitation performance of the optimized centrifugal pump is also better than the original pump. The results in this paper can provide references for parameter selection ($b_2$, $\beta_2$, $\varphi$) in the centrifugal pump design.

**Keywords:** high specific speed centrifugal pump; performance optimization; orthogonal design method

## 1. Introduction

High specific speed centrifugal pumps with low head and large flow are generally used in rural irrigation, urban drainage, fish pond aquaculture, aerospace engineering, etc. As the internal flow during the operation of centrifugal pump is extremely complicated, especially under off-design working conditions, it is often accompanied by unstable phenomena, such as overload operation, cavitation, and vibration, which will affect the normal operation. Therefore, it is particularly important to optimize the performance to improve their stability and reliability (Zhang et al. [1]). Numerical simulation has become a common research method with the development of computational fluid dynamics (CFD) (Wei et al. [2]; Zhou et al. [3]; Han et al. [4]).

Generally, there are two ways to optimize the performance of centrifugal pump. One is to study the effect of a single parameter or structure on the performance of centrifugal pump. Skrzypacz and Bieganowski [5] studied the influence of micro grooves on centrifugal pump performance by numerical simulation and experiment. The results illustrated that micro-blades can make the velocity distribution in the impeller passage more uniform, which thus improves the head and efficiency of the pump. Chen, He and Liu [6] put forward the concept of twisted vice blade, and verified that the use of twisted vice blades can effectively improve the comprehensive performance of centrifugal pumps. Nishi, Fukutomi, and Fujiwara [7] found the influence of blade outlet angle on radial thrust and modeled components

is obvious by experiments and CFD analysis. Fu, Zhu, Jiang, and Li [8] uncovered how the diffuser vane height affects the pump performance. They concluded that reducing the diffuser vane height could improve the output work of impeller. Cui, Wang, Zhu, and Jin [9] found that a larger blade outlet angle could improve the internal flow of low-specific-speed centrifugal pump and improve its working efficiency.

The other common optimization method is to use some mathematical models or algorithms to study the comprehensive effects of several parameters on the performance of centrifugal pumps, such as DOE (Design of Experiments), approximation models, genetic algorithm, etc. (Zhou et al. [10]; Lyn et al. [11]; Nasruddin et al. [12]). Wang, Feng, Ye, and Luo, Liu [13] selected blade inlet angle, outlet angle, and blade wrap angle as the optimization variables. The multi-objective optimization was carried out based on NSGA-II genetic algorithm. The optimization results show that the algorithm can effectively improve the performance of centrifugal slurry pump. Derakhshan, Pourmahdavi, Abdolahnejad, Reihani, and Ojaghi [14] adopted the ABC (artificial bee colony) and ANNs (artificial neural networks) algorithm to design a new flow passage shape of centrifugal pump. Zhang, Hu, Wu, Zhang, and Chen [15] used Kriging metamodels to optimize double suction centrifugal pump based on four different parameters. Lomakin, Chaburk, and Kuleshova [16] chose six parameters of impeller and guide vane as optimization parameters, and used LP-tau algorithm to optimize the centrifugal pump to improve its comprehensive performance, including increasing head, reducing cavitation, and vibration phenomenon. Koor, Vassiljev and Koppel [17] used the LMA (Levenberg-Marquardt algorithm) to maximize the total efficiency of the pump system and thereby minimize energy consumption.

Among these optimization methods, the orthogonal design experiment in DOE is a more efficient and economical method. Pei, Yin, Yuan, and Wang [18] used the orthogonal design experiment to study the influence of impeller geometric parameters on the cavitation of the pump, and found that the optimum combination of parameters had a better anti-cavitation and hydraulic performance than the original one. Huang and Liu [19] studied the influence of four parameters on centrifugal pump by orthogonal experiment, which provided reference for the design of rotor and volute. Wang and Huo [20] concluded that the optimized pump had no obvious "jet-wake" phenomenon by an orthogonal test. The blade outlet angle and blade leading edge position of impeller are critical factors affecting efficiency and anti-cavitation performance. Although orthogonal design experiments have been used to optimize centrifugal pumps in some references, few references have been made to optimize cavitation performance, and there is basically no research regarding centrifugal pumps with high specific speed. In this paper, the orthogonal design method is adopted to optimize the performance of high specific speed centrifugal pump basing on three parameters, namely, the blade outlet width $b_2$, the blade outlet angle $\beta_2$, and the blade wrap angle $\varphi$. First, the hydraulic experiment of the prototype pump is carried out to verify that the three-dimensional (3D) model can be simulated. Subsequently, nine groups of representative parameter combinations are obtained according to the orthogonal table. The number of simulations are greatly reduced. Finally, the best combination of parameters is obtained through numerical simulation results. For the high specific speed centrifugal pump that was studied in this paper, the hydraulic performance and anti-cavitation performance can be improved by properly increasing the blade outlet width, reducing the blade outlet angel, and blade wrap angle.

## 2. Calculation Model and Method

### 2.1. Calculation Model

The calculation model in this paper is a single stage single suction centrifugal pump with a high specific speed of 51 ($n_s = \frac{n \sqrt{Q}}{H^{3/4}}$). The design parameters are as follows: flow rate $Q = 100$ m$^3$/h, head $H = 20$ m, rotational speed $n = 2900$ r/min. Table 1 shows other main geometric parameters of the model.

**Table 1.** Main parameters of centrifugal pump impeller.

| Parameter | Value |
|---|---|
| Impeller | |
| Suction diameter ($D_j$) | 96 mm |
| Impeller diameter ($D_2$) | 135 mm |
| Outlet width ($b_2$) | 23 mm |
| Blade inlet angle ($\beta_1$) | 42° |
| Blade outlet angle ($\beta_2$) | 27° |
| Blade wrap angle ($\varphi$) | 120° |
| Blade number ($Z$) | 5 |
| Blade thickness ($\delta$) | 3 mm |
| Blade shape | Twist blade |
| Volute | |
| Base diameter ($D_3$) | 140 mm |
| Inlet width ($b_3$) | 46 mm |
| Tongue angle ($\varphi_0$) | 34° |
| Diffuser outlet diameter ($D_d$) | 125 mm |
| Diffuser length ($L$) | 265 mm |

Figure 1 shows the 3D model of pump established by software CFturbo (CFturbo 10, CFturbo GmbH, Dresden, Germany) and SolidWorks (SolidWorks 2017, Dassault Systèmes SolidWorks Corporation, Waltham, MA, US), which include four parts, namely inlet, impeller, volute, and outlet. The impeller inlet and the volute outlet are properly extended to reduce the influence of water backflow.

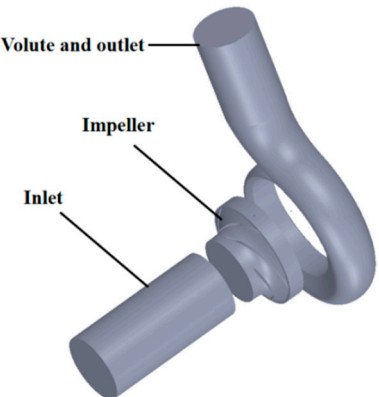

**Figure 1.** The 3D model of centrifugal pump.

*2.2. Governing Equation and Turbulence Model*

The Navier–Stokes governing equations consisting of momentum conservation equation and energy conservation equation are used in the numerical simulation (Cohen and Eyal [21]). They can be expressed as:

$$\frac{\partial u}{\partial t} = -(u \cdot \nabla)u + v\nabla^2 u - \frac{1}{\rho}\nabla P + F \tag{1}$$

$$\nabla \cdot u = 0 \tag{2}$$

where $u$ is velocity, $t$ is time; $\frac{\partial u}{\partial t}$ is the rate of velocity vs time; $\nabla$ is gradient operator; $v$ is viscosity coefficient; $P$ is pressure; $\rho$ is density; and, $F$ is external force.

The turbulence model chosen in this paper is the Renormalization-group (RNG) k-ε model. Table 2 shows that the numerical prediction of the RNG k-ε turbulence model is in good agreement with the experimental data at the design flow rate. Accordingly, the RNG k-ε model is adopted in this paper.

**Table 2.** Selection of turbulence model.

| Turbulence Model | Head (m) | Efficiency (%) | Head Error (%) | Efficiency Error (%) |
|---|---|---|---|---|
| standard k-ε | 20.0726 | 83.09 | 2.65 | 2.44 |
| RNG k-ε | 19.9939 | 82.26 | 2.28 | 1.41 |
| realizable k-ε | 20.1325 | 83.41 | 2.96 | 2.83 |
| Experiment | 19.5538 | 81.11 | | |

The RNG k-ε model is a mathematical model that is derived from instantaneous N-S equation by using the "renormalization group method" (Coutier et al. [22]; Smith [23]). It can capture turbulent diffusion at multiple scales.

*2.3. Cavitation Model*

The cavitation model is based on the Rayleigh–Plesset equation, which is provided by ANSYS CFX software (ANSYS 15.0, ANSYS Inc., Canonsburg, PA, USA). The cavitation flow is usually regarded as a single-phase flow with average fluid characteristics. It is used to calculate the mass transfer process between the gas phase and liquid phase (Singhal et al. [24]).

$$R_b \frac{d^2 R_B}{dt^2} + \frac{3}{2}\left(\frac{dR_B}{dt}\right)^2 + \frac{2S}{\rho_1 R_B} = \frac{P_{sat} - P}{\rho_1} \tag{3}$$

where $R_B$ is bubble radius, $P_{sat}$ is saturated vapor pressure, and $\rho_1$ is liquid density.

Ignoring the surface tension and the second-order term of the bubble during vaporization and condensation, the growth rate of the bubble can be obtained from Equation (9).

$$\frac{dR_B}{dt} = \sqrt{\frac{2(P_{sat} - P)}{3\rho_1}} \tag{4}$$

*2.4. Grid Independence Analysis*

The computational grids are generated by using ICEM CFD (ICEM CFD 15.0, ANSYS Inc., Canonsburg, PA, USA). Aiming at the complex flow passage structure of centrifugal pump, unstructured tetrahedral mesh with strong adaptability is selected for mesh generation (Ahn and Kwon [25]). The local refinement is carried out at the surface of blades and the tongue of volute to deal with large flow gradient change. Figure 2 shows the computational domain grids of model pump.

The grid independence verification is required in order to reduce the influence of grid number on simulation results. Table 3 shows the simulation head and efficiency under different grid numbers. From the table, we can find that the error of head and efficiency is less than 2% between scheme II and III, so the influence of grid number on calculation can be neglected. Scheme II is finally selected for numerical simulation after considering the computational cost and grid independence.

**Table 3.** Grid Independence Verification.

| Scheme | Grid Number | Head/m | Efficiency/% |
|---|---|---|---|
| I | 869,171 | 20.48 | 83.18 |
| II | 1,330,820 | 19.99 | 82.26 |
| III | 2,102,155 | 19.72 | 81.79 |

The global element scale factor of scheme II is set to 1 and the max element of global element seed size is set to 4. The minimum grid quality is bigger than 0.3 and the minimum angle is bigger than 14°. The grid quality is satisfied. The $y^+$ values of each part ranged from 30 to 100 (Zhou, Shi, Lu, Hu and Wu [26]).

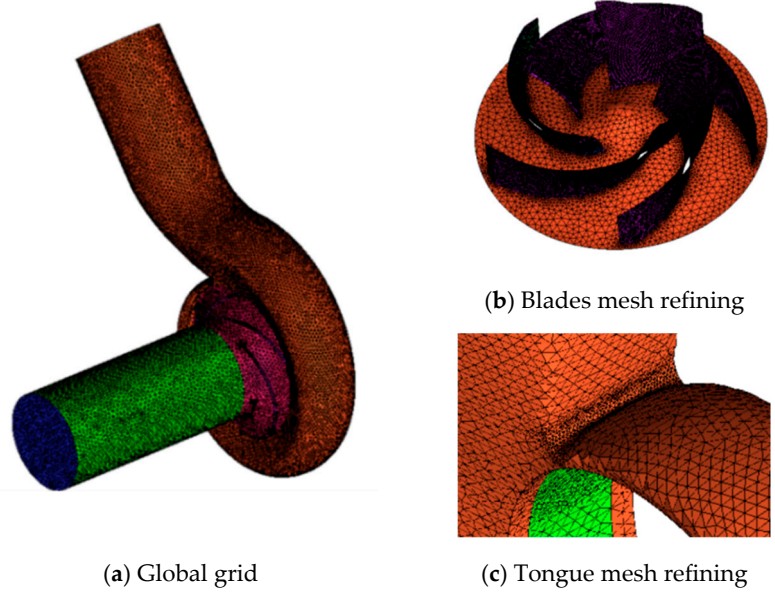

(**b**) Blades mesh refining

(**a**) Global grid

(**c**) Tongue mesh refining

**Figure 2.** Computational domain grid.

## 2.5. Calculation Method

The simulation in this paper is carried out in the software ANSYS-CFX. The RNG k-ε turbulence model is used to close the N-S equations and the standard wall function method is used to deal with the near-wall region. The SIMPLE algorithm is used for the coupling of velocity and pressure. The second-order upwind method is used for the discretization of momentum, turbulent kinetic energy, and dissipation rate equations. The impeller region is set as the rotating domain and the rest region are set as stationary domain. Frozen rotor method is used to connect the dynamic-static interface. Turbulence intensity is set to medium (intensity = 5%). Total pressure (1 atm) is used as the inlet boundary condition and mass flow rate (27.78 kg/s) is used as the outlet boundary condition. The boundary conditions at the wall of hub, shroud, outlet, inlet, and volute are set to no-slip condition. In solver control, the maximum iteration step is set to 1000 and the residual target of convergence criteria is set to $10^{-5}$ (Babayigit et al. [27]). Table 4 shows all of the boundary conditions settings.

**Table 4.** Boundary condition setting.

| Position | Boundary Condition |
| --- | --- |
| Inlet | Total pressure (1 atm) |
| Outlet | Mass flow rate (27.78 kg/s) |
| Walls | No-slip wall |
| Impeller | Rotating domain |
| Volute | Stationary domain |
| Interface | General connection |
| Near-wall region | Standard wall function |

The cavitation model is based on the Rayleigh–Plesset equation. The single-phase calculation results without cavitation are taken as the initial values of the two-phase flow of cavitation in order to improve the convergence rate of the calculation. The internal cavitation of the pump is realized by gradually reducing the inlet total pressure. The volume of the vapour at the inlet is set to be 0 and the volume fraction of water is set as 1. For fluid pair models, the mean diameter of bubbles is set to $2 \times 10^{-6}$, the saturation pressure is set to 3326 Pa, the cavitation condensation coefficient is set to 0.01, and the cavitation vaporization coefficient is set to 50.

## 2.6. Prediction Algorithm

ANSYS-CFX simulation cannot directly obtain parameters, such as head and efficiency of centrifugal pump, so it needs to input formulas for calculation.

The calculation formula of centrifugal pump head can be expressed as (Ding et al. [28]):

$$H = \frac{p_{out} - p_{in}}{\rho g} \tag{5}$$

where $H$ is the centrifugal pump head, $p_{out}$ is the volute outlet pressure, $p_{in}$ is the impeller inlet pressure, $\rho$ is the fluid density, and $g$ is the gravity acceleration.

The total efficiency ($\eta$) is calculated, as follows:

$$\eta = \left( \frac{1}{\eta_v \eta_h} + \frac{\Delta p_d}{p_e} + 0.03 \right)^{-1} \tag{6}$$

where $\eta_v$ is the volume efficiency, $\eta_h$ is the flow efficiency, $\Delta p_d$ is the friction loss of impeller disk, and $p_e$ is the effective power of fluid.

## 2.7. Model Verification

The characteristic curve of the model pump is measured on the experimental platform in order to verify that the model can simulate correctly, as shown in Figure 3. Pressure meters are installed in the inlet and outlet pipe of centrifugal pump. Before starting the centrifugal pump, the pump and the inlet pipe should be filled with water to discharge the air in the pump.

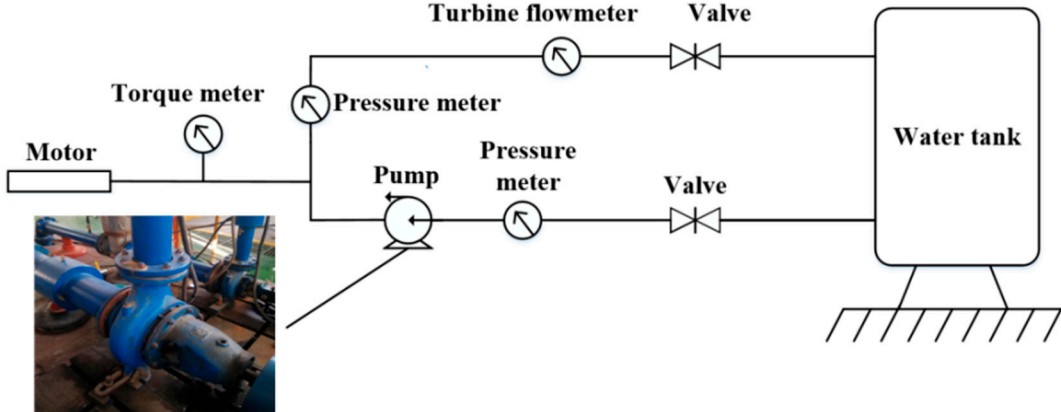

**Figure 3.** Hydraulic performance test system.

Table 5 shows all of the equipment and parameters of the experiment. The centrifugal pump is driven by Y160M1-2 motor, its rated voltage is 380 V, and rated power is 11 kw.

**Table 5.** Instruments for experiment.

| Measured Parameter | Equipment Name | Model | Minimum Scale | Precision |
|---|---|---|---|---|
| Pressure | Pressure meter | YM3 | 100 Pa | ±0.1 |
| Flow rate | Turbine meter | LWGY-MIK | 0.01 Pa | ±0.5 |
| Test speed | | | 0.1 r/min | |
| Shaft power | Torque meter | YXS-NJ | 0.1 kw | ±0.5 |
| Torque | | | 0.1 N m | |

In the process of experiment, uncertainty will arise due to the accuracy of measuring equipment, artificial operation errors, and experimental environment. In each experiment, the uncertainty of hydraulic efficiency that is caused by different variables can be expressed by the following formula (Babayigit et al. [29]).

$$\omega_{\eta_h} = \pm 100 \left[ \left( \frac{\partial \eta_h}{\partial p} \omega_p \right)^2 + \left( \frac{\partial \eta_h}{\partial \dot{Q}} \omega_{\dot{Q}} \right)^2 + \left( \frac{\partial \eta_h}{\partial N} \omega_N \right)^2 \right]^{0.5} / \eta_h \tag{7}$$

where $\eta_h$ is flow efficiency. (Ding et al. [28]). $\omega_p$, $\omega_{\dot{Q}}$, and $\omega_N$ represent the precision of turbine meter, pressure meters, and torque meter, respectively. Precision values of the measurement parameters for pressure (Pa), flow rate (m³/h), and torque are shown in Table 5. The variation of uncertainty value again flow rate is shown in Figure 4. During the experiment, the uncertainty value is less than ±1%. The uncertainty value continues to decline, as the flow rate of change affects the calculation of hydraulic efficiency more than other parameters.

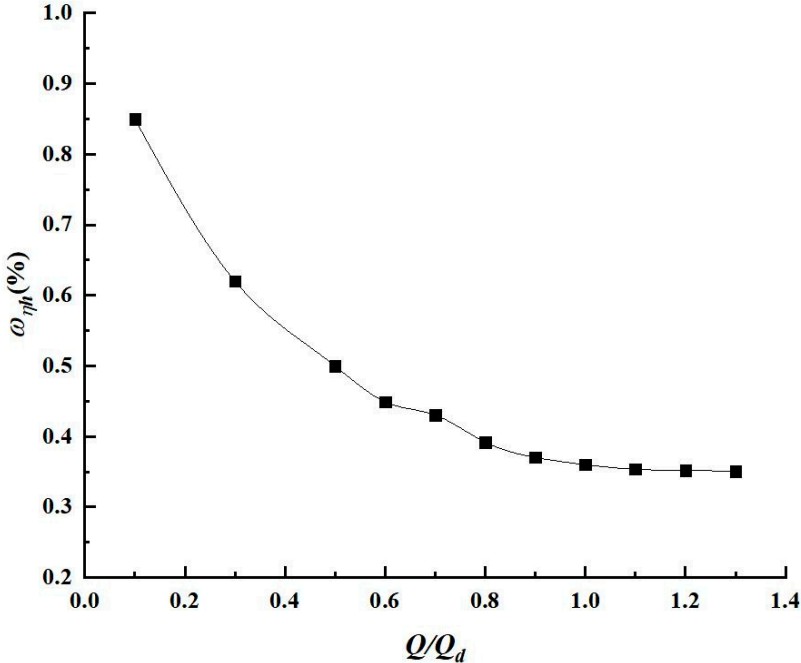

**Figure 4.** Variation of uncertainty value with flow rate.

Figure 5 shows the characteristic curves of simulation and experiment. It can be seen that the simulated head and efficiency characteristic curves are consistent with the experimental curves. As the simulation does not take into account many losses, the values of head and efficiency are always slightly higher than the experimental values. The maximum difference is less than 5% for the head and efficiency values of simulation and experiment under different flow rate, so the model used in this paper can be correctly simulated.

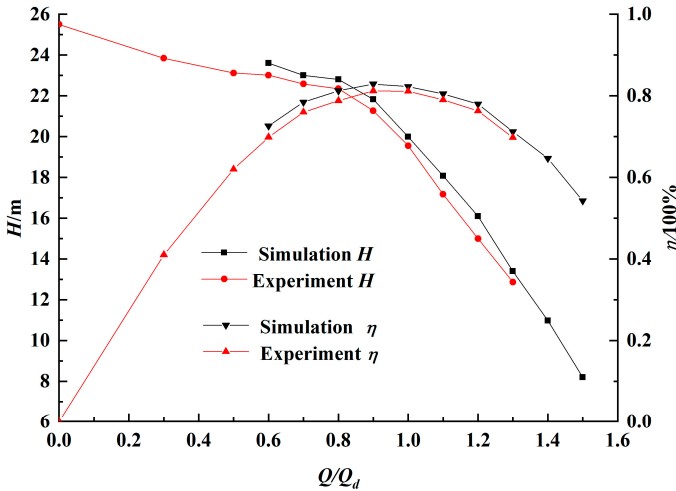

**Figure 5.** Characteristic curves of simulation and experiment.

## 3. Orthogonal Experimental Scheme

The main purpose of this paper is to optimize the centrifugal pump by orthogonal experiment design. Orthogonal experiment design is a method to analyze the influence of many factors on the whole situation by using the orthogonal Table. The principle of this method is to select some representative horizontal combinations from all the combinations of experimental factors through orthogonal relations and simulate the experimental results of these combinations. Ultimately, we can obtain the influence of each parameter on the overall performance and find the optimal horizontal combination. It is an efficient and convenient optimization design method.

### 3.1. Factors Selection

High specific speed centrifugal pump has wide channel and the blade wrap angle size will directly affect its hydraulic performance (Tan et al. [30]). The blade wrap angle represents the diffusion degree of blade passage. Figure 6 shows the schematic diagram of blade wrap angle; it is defined as the included angle of two lines, one is the connecting line of the blade inlet edge and the centre, another is the connecting line of blade outlet edge and the center.

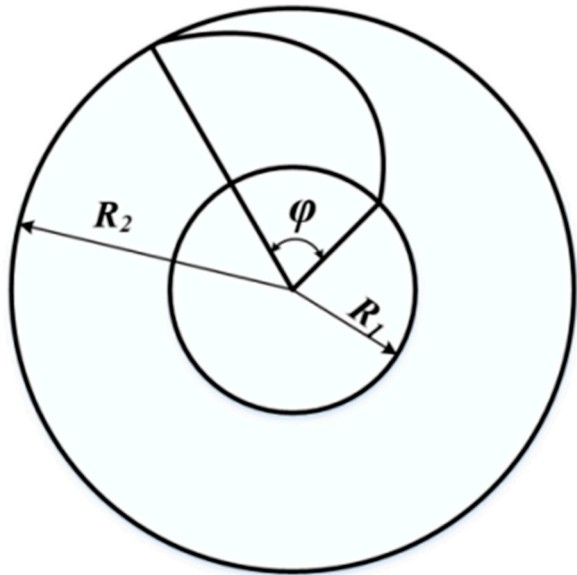

**Figure 6.** Schematic diagram of blade wrap angle.

Table 6 shows the optimal value of $Z\varphi/360$ corresponding to specific speed $n_s$. According to the number of blades (Z = 5) and the specific speed of centrifugal pump ($n_s$ = 51), the warp angle range is calculated to be 108–129.6. Therefore, three different blade wrap angles of 112°, 120°, and 128° are selected in this study.

**Table 6.** Selection of wrap angle.

| $n_s$ | 10–14 | 14–19 | 22–33 | 36–60 | 63–77 |
|---|---|---|---|---|---|
| $Z\varphi/360$ | 2.1–2.3 | 1.9–2.1 | 1.7–1.9 | 1.5–1.8 | 1.46–1.65 |

The blade outlet angles are selected as 24°, 27°, and 30° according to previous studies (Ding et al. [28]). The blade outlet widths are selected to 22 mm, 23 mm, and 24 mm.

*3.2. Orthogonal Table Design*

According to the principle of orthogonal design experiment method, the orthogonal tables of the above three parameters are established. Three different levels are selected for each factor, as shown in Table 7.

**Table 7.** Level of factors in orthogonal experiments.

| Level | Factor | | | |
|---|---|---|---|---|
| | A | B | C | D |
| | $b_2$/mm | $\beta_2$/(°) | $\varphi$/(°) | — |
| 1 | 23 | 27 | 120 | 1 |
| 2 | 22 | 24 | 112 | 2 |
| 3 | 24 | 30 | 128 | 3 |

According to $L_9$ ($3^4$) orthogonal table, nine groups of orthogonal test schemes are set up, as shown in Table 8. An experiment-independent factor is added to supplement the orthogonal table in the last column, as there are only $3^4$ orthogonal table in the orthogonal experimental and no $3^3$ table.

**Table 8.** The orthogonal table design.

| Scheme | Serial Number | | | | Corresponding Parameters | | | |
|---|---|---|---|---|---|---|---|---|
| | A | B | C | D | $b_2$/mm | $\beta_2$/(°) | $\varphi$/(°) | — |
| 1 | $A_1$ | $B_1$ | $C_1$ | $D_1$ | 23 | 27 | 120 | 1 |
| 2 | $A_1$ | $B_2$ | $C_2$ | $D_2$ | 23 | 24 | 112 | 2 |
| 3 | $A_1$ | $B_3$ | $C_3$ | $D_3$ | 23 | 30 | 128 | 3 |
| 4 | $A_2$ | $B_1$ | $C_2$ | $D_3$ | 22 | 27 | 112 | 3 |
| 5 | $A_2$ | $B_2$ | $C_3$ | $D_1$ | 22 | 24 | 128 | 1 |
| 6 | $A_2$ | $B_3$ | $C_1$ | $D_2$ | 22 | 30 | 120 | 2 |
| 7 | $A_3$ | $B_1$ | $C_3$ | $D_2$ | 24 | 27 | 128 | 2 |
| 8 | $A_3$ | $B_2$ | $C_1$ | $D_3$ | 24 | 24 | 120 | 3 |
| 9 | $A_3$ | $B_3$ | $C_2$ | $D_1$ | 24 | 30 | 112 | 1 |

*3.3. Orthogonal Analysis*

Through numerical simulation in ANSYS CFX, the head (Equation (5)) and efficiency (Equation (6)) of centrifugal pump for nine orthogonal design schemes are obtained, as shown in Table 9. The results can be used to find suitable parameters combination.

**Table 9.** Simulation results in orthogonal design.

| Scheme | $H/m$ | $\eta/100\%$ |
|--------|-------|--------------|
| 1 | 19.99 | 82.26 |
| 2 | 20.29 | 82.77 |
| 3 | 19.58 | 81.03 |
| 4 | 20.07 | 82.36 |
| 5 | 19.83 | 82.67 |
| 6 | 19.70 | 81.46 |
| 7 | 19.83 | 81.90 |
| 8 | 20.74 | 82.81 |
| 9 | 20.44 | 81.60 |

For the orthogonal design, range analysis is used to seek the optimal parameters combination based on the simulation results in Table 9, and the algorithm of range analysis can be expressed, as follows:

$$k_i = \frac{1}{3}K_i = \frac{1}{3}\sum_{j=1}^{N_i} y_{i,j} \tag{8}$$

$$R = \max(k_1, k_2, \dots, k_i) - \min(k_1, k_2, \dots, k_i) \tag{9}$$

where $K_i$ is the sum of $i$ levels of each factor, $k_i$ is the average value, and $R$ is the range, which reflects the influence degree of factors.

$K_i$, $k_i$, and $R$ are calculated by the above formula. The larger the $K_i$, the higher the head and efficiency at this level. The larger the $R$, the greater the influence of this factor on pump performance. Table 10 shows the results of range analysis.

**Table 10.** Range analysis of head.

| Index | Factor | | |
|-------|--------|--------|--------|
| | **A** | **B** | **C** |
| $K_1$ | 59.86 | 59.89 | 60.43 |
| $K_2$ | 59.6 | 60.86 | 60.8 |
| $K_3$ | 61.01 | 59.72 | 59.24 |
| $k_1$ | 19.95 | 19.96 | 20.14 |
| $k_2$ | 19.87 | 20.29 | 20.27 |
| $k_3$ | 20.34 | 19.91 | 19.75 |
| $R$ | 0.47 | 0.38 | 0.52 |

In Table 10, the influence order of factors on head is as follows: $\varphi$, $b_2$, and $\beta_2$, so the optimal combination for head is $A_3B_2C_2$. In Table 11, the influence order of factors on efficiency is: $\beta_2$, $\varphi$, and $b_2$, so the optimal combination of efficiency is $A_2B_2C_2$. Here, the highest efficiency of centrifugal pump is selected as the evaluation criterion, and the optimal combination is $A_3B_2C_2$. Therefore, the geometric parameters of the optimal impeller model are outlet width $b_2 = 24$ mm, blade outlet angle $\beta_2 = 24°$, and blade wrap angle $\varphi = 112°$, respectively.

**Table 11.** Variation in vapor volume fraction.

| Inlet Pressure (kPa) | Vapor Volume Fraction | |
| --- | --- | --- |
| | Origin | Optimal |
| 30.397 | 0.094769443 | 0.059625714 |
| 40.530 | 0.062375526 | 0.036388726 |
| 50.662 | 0.050064852 | 0.028880441 |
| 60.794 | 0.022115802 | 0.014879935 |
| 70.927 | 0.016335408 | 0.013927614 |
| 81.060 | 0.006443528 | 0.0005504 |
| 121.589 | 0 | 0 |

## 4. Results and Discussion

### 4.1. Numerical Simulation of Optimal Model

When compared with the original model, the blade outlet width of the optimized model increases by 1 mm, the blade outlet angle decreases by three degrees and the blade wrap angle decreases by eight degrees. Figure 7 gives a comparison between the optimal centrifugal pump blade and the original pump blade.

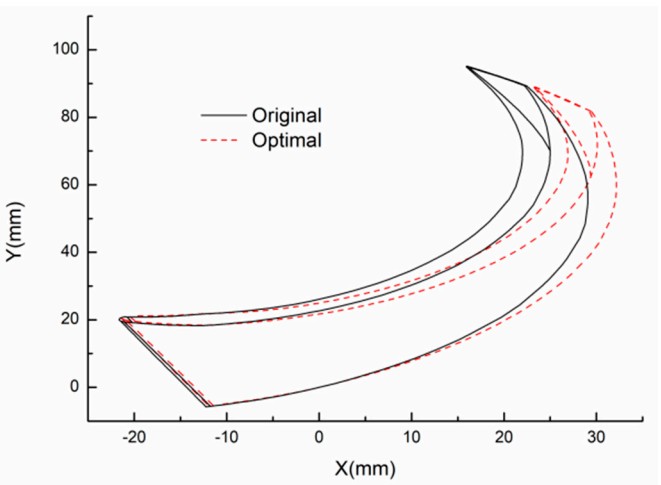

**Figure 7.** Optimal blade shape.

A new three-dimensional model of centrifugal pump is built and simulated according to the optimized parameters. Figure 8 shows the numerical simulation results of the external characteristics of centrifugal pumps before and after optimization.

From Figure 8a, it can be seen that the head of the optimal pump is higher than the original pump under all working conditions. At low flow rate (0.6 $Q_d$–1.0 $Q_d$), the head increment of optimal pump is small, but it becomes large with the increase of flow rate. At the design flow rate, the head of original and optimal pump are 19.99 m, 20.74 m, respectively, and the head after optimization has been increased by 3.75%. Under 1.5 times design flow rate, the head of the original and optimal pump are 8.19 m and 9.17 m, respectively, the head increment reaches to 12%. As can be seen from Figure 8b, the trend of efficiency change is different from that of head change. In the low flow rate region (0.6 $Q_d$–1.0 $Q_d$), the efficiency of the original pump is higher than the optimal pump. At 0.6 $Q_d$, the efficiency of the original pump is 3.7% higher than that of the optimal pump. However, in the high flow rate region (1.0 $Q_d$–1.5 $Q_d$), the efficiency of the optimal pump is obviously higher than the original pump and the value increases by 6.9% at 1.5 $Q_d$.

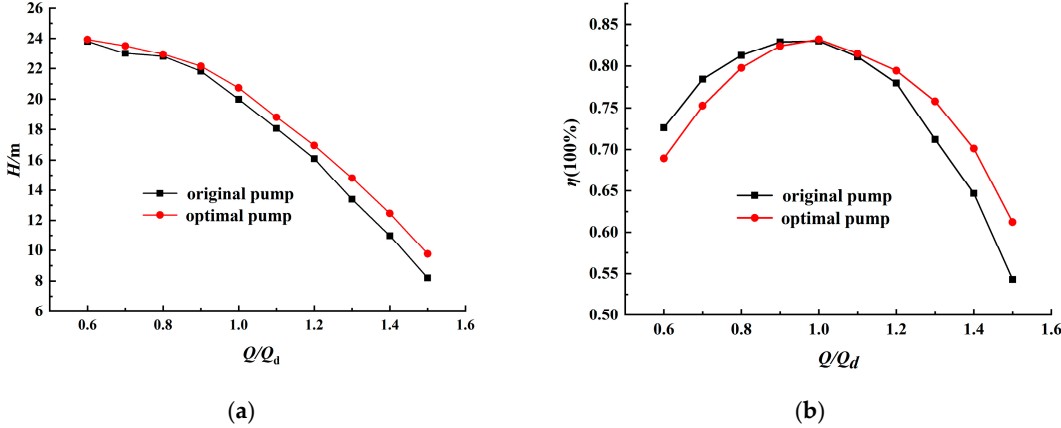

**Figure 8.** Head and efficiency of original pump and optimal pump. (**a**) *Q-H*; (**b**) *Q-η*.

As shown in Figure 9, with the decrease of NPSH$_a$ (available net positive suction head), the change of head is very small. When NPSH$_a$ decreases to a certain value, the head begins to decline faster. NPSH$_r$ (require net positive suction head) corresponds to that when head drops by 3%. The larger NPSH$_r$ means the greater the pressure drops, and the worse the anti-cavitation performance of the pump. The NPSH$_r$ for the optimal pump is 5.38 m, which is smaller than that of 5.92 m for the original pump, achieving a appropriate improvement on the anti-cavitation performance.

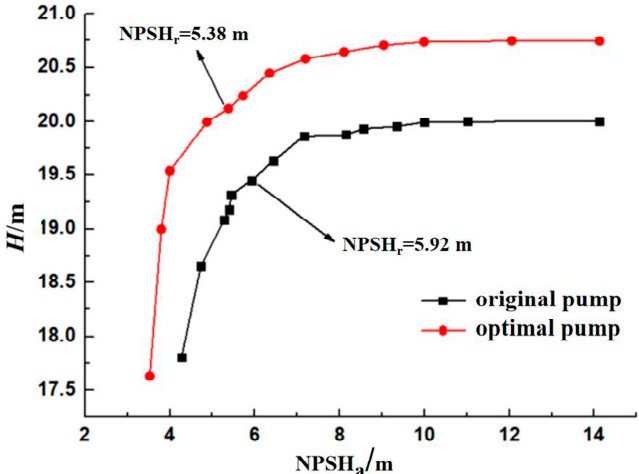

**Figure 9.** Cavitation performance of original pump and optimal pump.

*4.2. Internal Flow Analysis*

By comparing the internal flow characteristics of the original pump and the optimal pump, the reasons for the influence of three factors on the performance of high specific speed centrifugal pump are found. Figure 10 shows the total pressure distributions under 0.6 $Q_d$, 1.0 $Q_d$, and 1.5 $Q_d$, respectively. It can be seen that the pressure reaches its lowest value at the suction side near the leading edge, where cavitation might begin to appear. The pressure increases along with the passage and reaches the maximum at the trailing edge. In addition, the pressure distribution at the trailing edge of the original impeller is less uniform than that of the optimal one. The optimal pump has a smaller minimum pressure area, which means less possibility of cavitation, especially in high flow rate. As shown in Figure 10e,f, optimization, the low pressure area (represented by the numbers I and II) of the pump is obviously reduced after optimization.

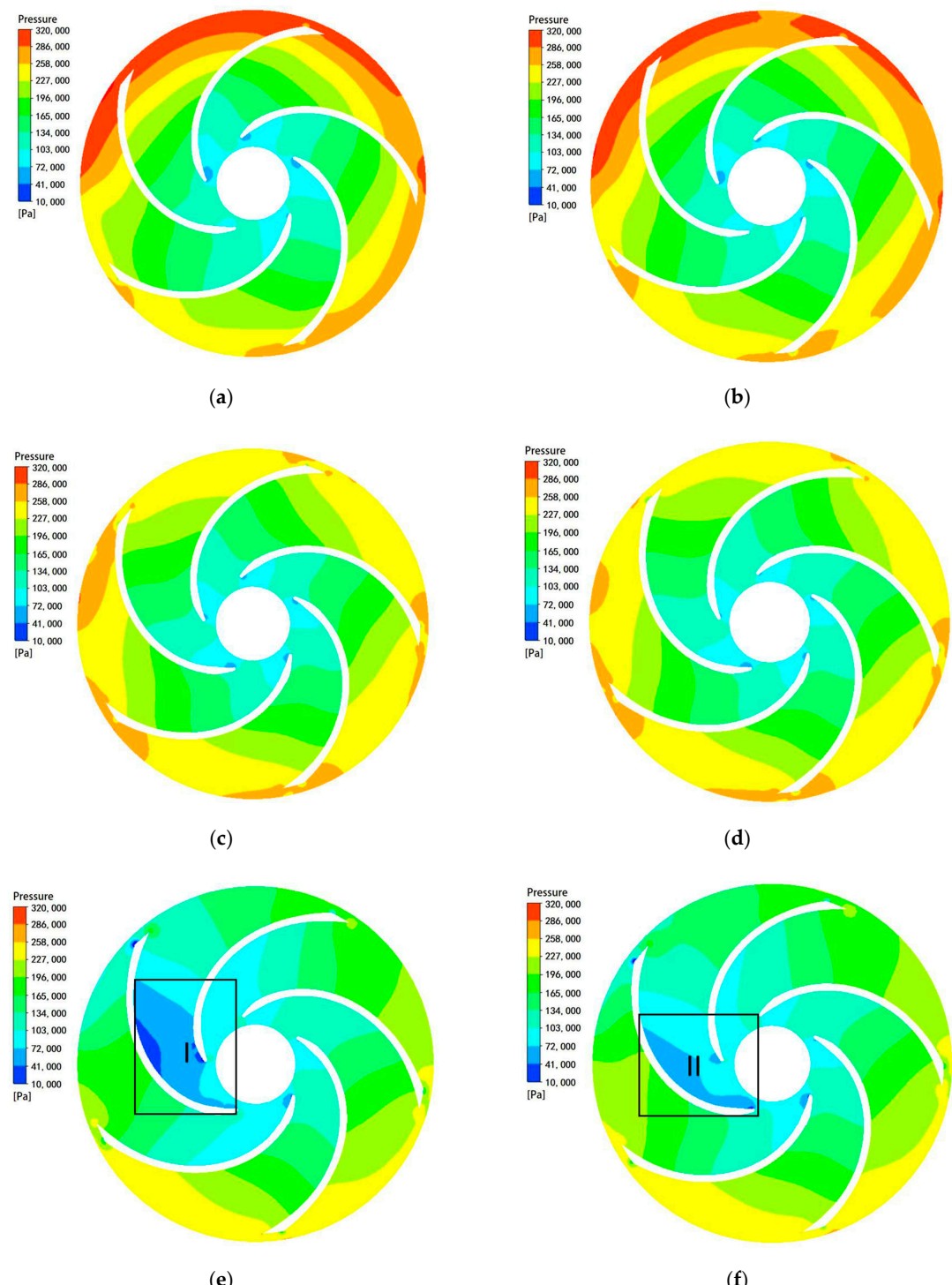

**Figure 10.** Pressure distributions in impeller of original and optimal pump. (**a**) 0.6 $Q_d$ (original); (**b**) 0.6 $Q_d$ (optimal); (**c**) 0.8 $Q_d$ (original); (**d**) 0.8 $Q_d$ (optimal); (**e**) 1.0 $Q_d$ (original); (**f**) 1.0 $Q_d$ (optimal).

Figure 11 shows the relative velocity distributions of the original and optimal pumps under three different flow rates. It can be seen that the velocity vector distribution in each channel is smoother after optimization, the internal vortex of impeller is reduced, and the flow state is obviously improved. The low velocity area in the optimal impeller is smaller than that in the original one, and this will reduce hydraulic loss and improve efficiency.

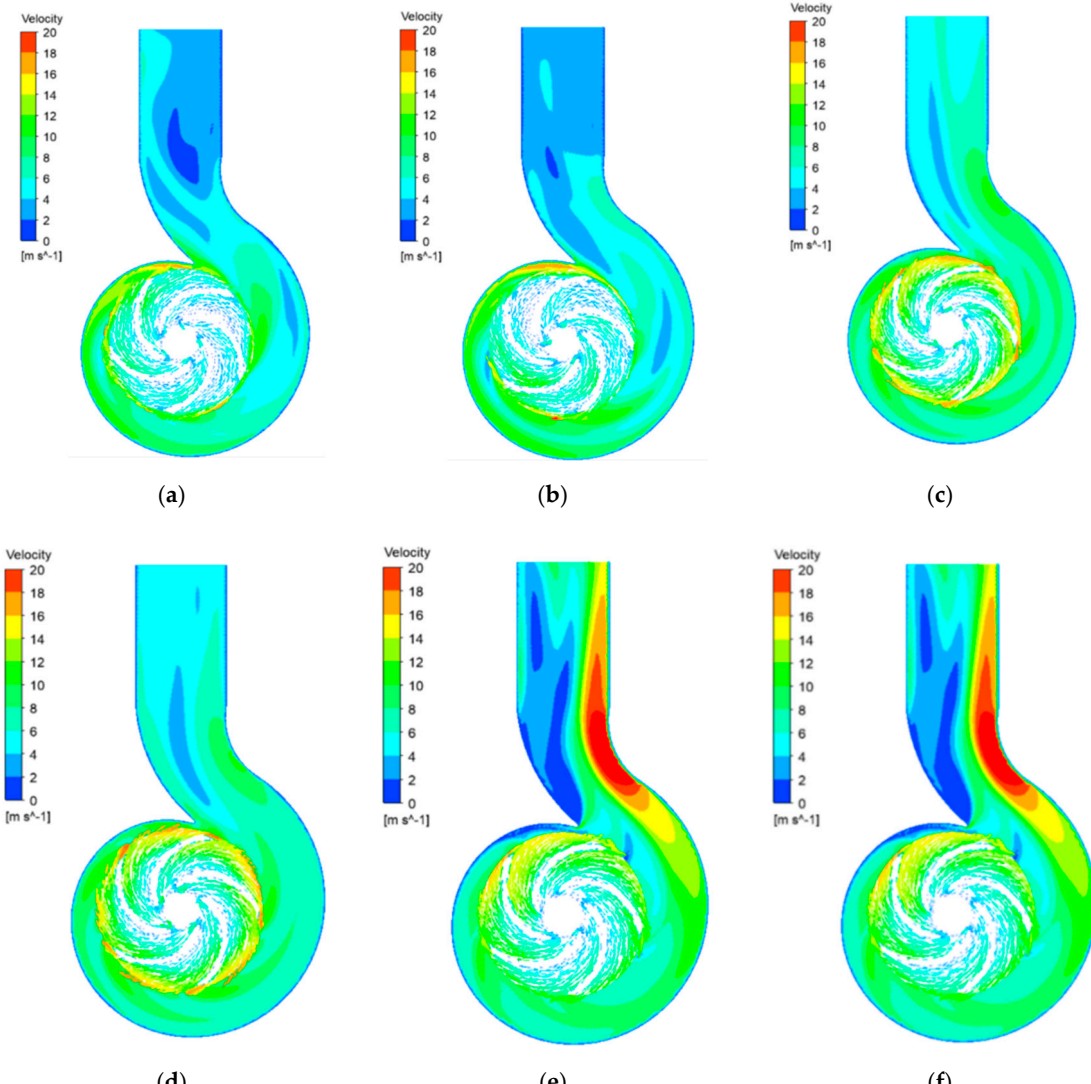

**Figure 11.** Velocity distributions in volute and vector distribution in impeller midspan of original pump and optimal pump. (**a**) 0.6 $Q_d$ (original); (**b**) 0.6 $Q_d$ (optimal); (**c**) 0.8 $Q_d$ (original); (**d**) 0.8 $Q_d$ (optimal); (**e**) 1.0 $Q_d$ (original); (**f**) 1.0 $Q_d$ (optimal).

Figure 12 shows the turbulent kinetic energy distribution in blade-to-blade view at 0.8 span of original pump and optimal pump. Turbulent kinetic energy refers to the part of energy that converts mechanical energy into heat energy in the fluid movement. The magnitude of turbulent kinetic energy is proportional to the intensity of turbulent vortices. $l_{span}$ is defined as the dimensionless distance from the hub to shroud of impeller, which ranges from 0 to 1. At low flow rate, the turbulent kinetic energy is not significantly reduced after optimization, and even some larger turbulent kinetic energy appears at the inlet. With the increase of the flow rate, the more violent the turbulent flow energy changes in the impeller passage, as shown in Figure 12b,c. Under the condition of large flow rate, the strong turbulent energy region in the impeller mainly concentrates at the inlet of the impeller and the turbulent energy fluctuation at the outlet of the impeller is relatively weak, as shown in Figure 12e,f. This is because, as the flow rate increases, the pressure difference between the impeller outlet and inlet decreases. The turbulent energy of the optimized impeller changes more smoothly, which indicates that the turbulent fluctuation of the optimal impeller is weaker and the hydraulic performance is better.

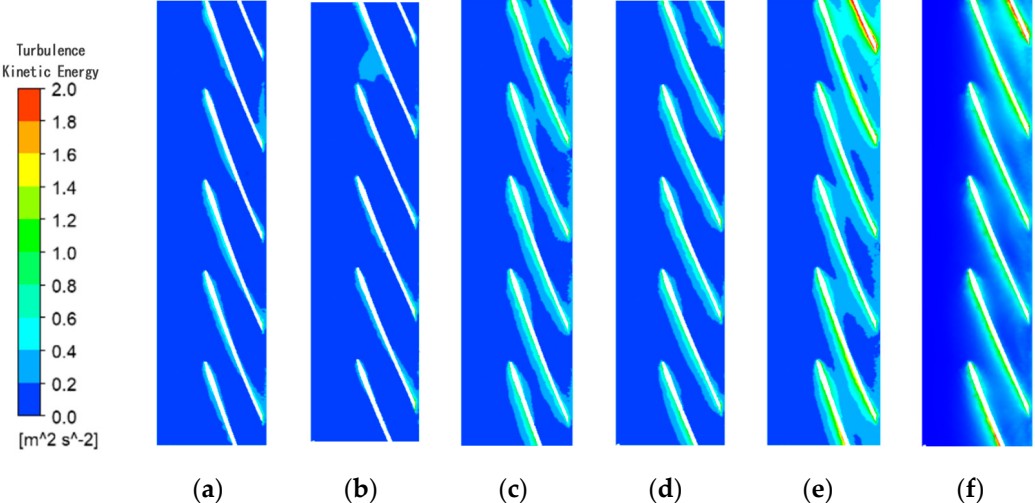

(a)      (b)      (c)      (d)      (e)      (f)

**Figure 12.** Turbulence kinetic energy distributions in blade-to-blade view at 0.8 $l_{span}$ of original pump and optimal pump. (**a**) original impeller at 0.6 $Q_d$, (**b**) optimal impeller at 0.6 $Q_d$, (**c**) original impeller at 1.0 $Q_d$, (**d**) optimal impeller at 1.0 $Q_d$, (**e**) optimal impeller at 1.5 $Q_d$, and (**f**) original impeller at 1.5 $Q_d$.

Figure 13 shows the vapor volume fraction distribution in impeller of original pump and optimal pump under different inlet pressure. As the inlet pressure gradually decreases to the saturated vapor pressure of water at 25 °C, vapor bubbles begin to appear in the channel of the centrifugal pump. It means that the cavitation phenomenon appears. When the inlet pressure of the pump dropped to 60.8 kPa, critical cavitation occurred in the original impeller. However, as can be seen from Figure 13a,b, the vapor volume fraction of optimal impeller is lower than the original impeller at the same condition, which means that the optimal impeller provided better anti-cavitation performance. When the inlet pressure is further reduced to 50.7 kPa, the vapor volume fraction is larger, which means that the cavitation becomes more serious. However, the vapor volume fraction (represented by numbers I and II) of the optimal pump is still significantly less than the original pump, as shown in Figure 13c,d. Figure 13e,f also show the same situation. The optimized centrifugal pump reduces the possibility of cavitation occurring.

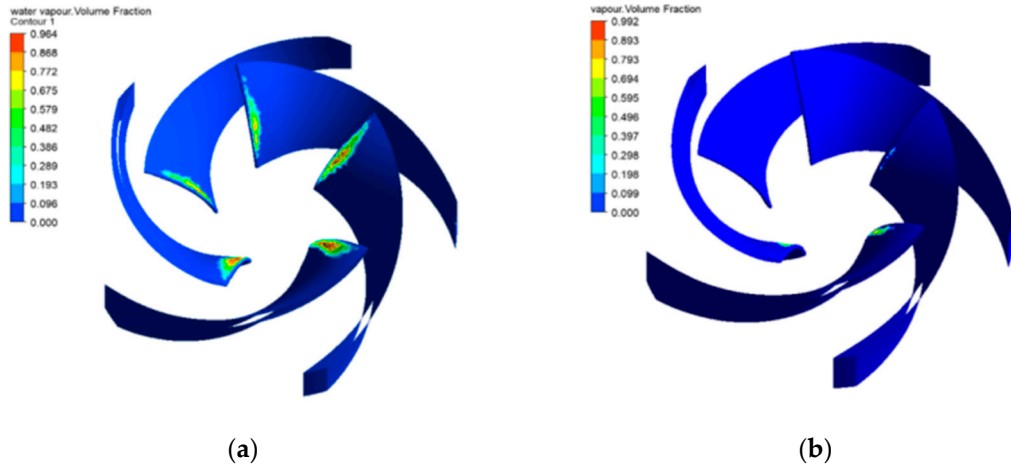

(a)                                     (b)

**Figure 13.** *Cont.*

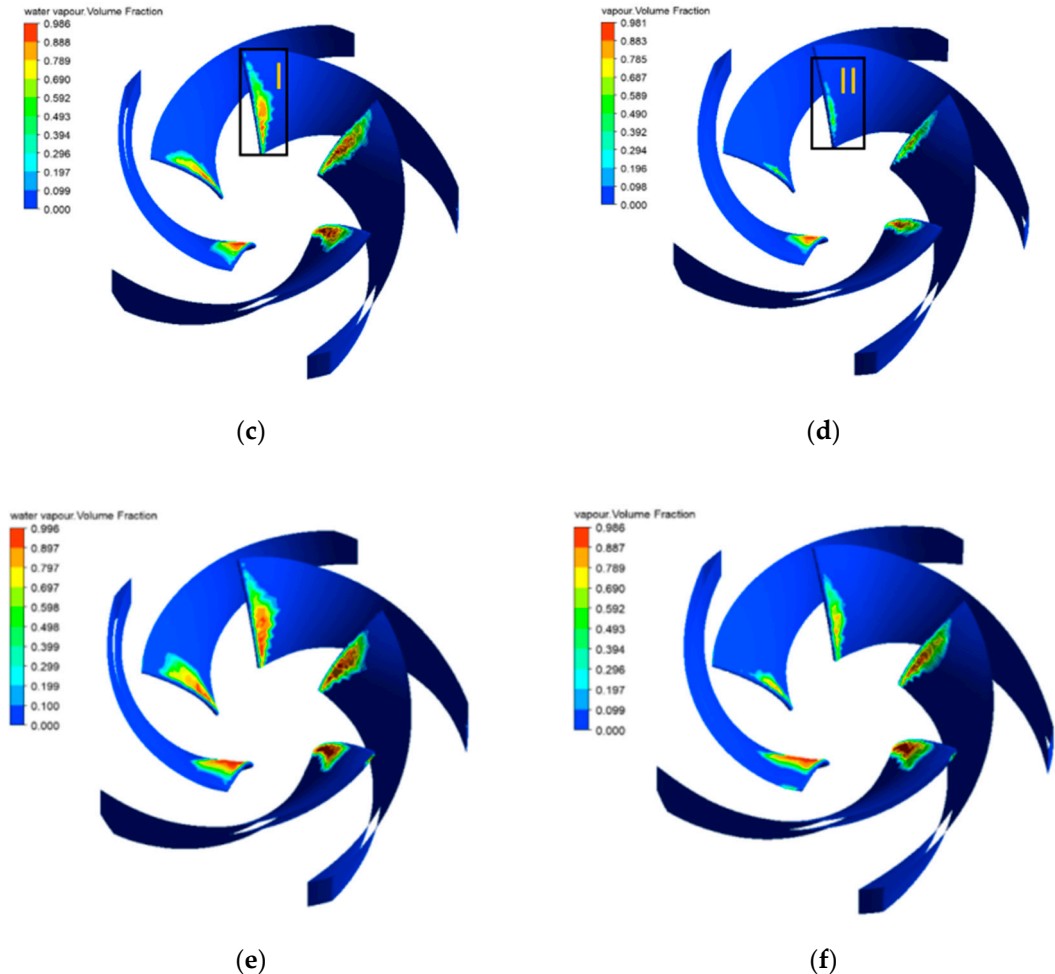

**Figure 13.** Vapor volume fraction distribution in impeller of original pump and optimal pump at different inlet pressure. (**a**) 60.8 kpa (original); (**b**) 60.8 kpa (optimal); (**c**) 50.7 kpa (original); (**d**) 50.7 kpa (optimal); (**e**) 40.5 kpa (original); (**f**) 40.5 kpa (optimal).

Table 11 shows the specific vapor volume fraction under different inlet pressure. When combining with Figure 13, it can be seen more intuitively that the optimized centrifugal pump has less vapor volume fraction and better anti-cavitation performance under the same condition.

## 5. Conclusions

In this paper, three impeller parameters are selected for orthogonal design experiments, and the optimal combination of parameters is obtained according to the results of numerical simulation. The main conclusions are as follows:

(1) Orthogonal design experiments can be used to study the influence of impeller parameters on the head and efficiency of centrifugal pumps. Among the three parameters studied, the wrap angle φ has the greatest impact on the head, while the blade outlet angle $\beta_2$ has the greatest impact on the efficiency.

(2) The head of optimal pump is obviously higher than that of the original pump, which increases by 0.74 m under the design flow rate and 1.58 m under 1.5 $Q_d$. However, the optimal pump's efficiency is lower than the original pump when the flow rate is less than 1.0 $Q_d$. When the flow rate is greater than 1.0 $Q_d$, the efficiency significantly increases; its value increases by 6.9% at 1.5 $Q_d$.

(3) The total pressure distribution of optimal pump is improved under the three operation points, especially at 1.5 $Q_d$. At the same time, the low velocity area of optimal pump is reduced, which will

cause less hydraulic loss and increase efficiency. Additionally, the optimized pump has less turbulent kinetic energy loss and better flow characteristics.

(4) The NPSH$_r$ value of optimized centrifugal pump is 0.54 m lower than the original model and its anti-cavitation performance is better.

The major optimizing objective in this paper is to improve efficiency. On this basis, the cavitation performance is also compared. However, the selected parameters are not enough to improve the cavitation performance, and some inlet parameters should be selected. In future work, the influence of volute parameters on the unsteady performance of centrifugal pump can also be studied. The optimization method can also adopt more advanced methods, such as Optimal Latin hypercube design.

**Author Contributions:** H.D. and Z.L. conceived and designed the content structure and experiments; Y.J. performed the experiments; Z.L. analyzed the data; Z.L. and X.S. did the simulation; H.D. and Z.L. wrote the paper.

**Funding:** This research was funded by National Natural Science Foundation of China (61803235), China Postdoctoral Science Foundation (2015M582112), key research and development project of Shandong province (2017GGX203005). The supports are gratefully acknowledged.

**Conflicts of Interest:** The authors declare no conflict of interest, financial or otherwise.

## Symbols

| | |
|---|---|
| $b_2$ | Blade outlet width |
| $b_3$ | Volute inlet width |
| $D_2$ | Impeller diameter |
| $D_d$ | Diffuser outlet diameter |
| $D_3$ | Volute base diameter |
| $D_j$ | Suction diameter |
| $F$ | External force |
| $g$ | gravity acceleration |
| $H$ | Head |
| $i$ | Factors |
| $k$ | Turbulent energy |
| $k_i$ | The average of $K_i$ |
| $K_i$ | The sum of $i$ levels of each factor |
| $L$ | Diffuser length |
| $n$ | Rotational speed |
| $n_s$ | Specific speed |
| $p$ | Pressure |
| $p_e$ | Effective power of fluid |
| $p_{in}$ | Impeller inlet pressure |
| $p_{out}$ | Volute outlet pressure |
| $Q$ | Flow rate |
| $Q_d$ | Design flow rate |
| $R$ | Range |
| $R_B$ | Bubble radius |
| $t$ | Time |
| $u$ | velocity |
| $v$ | viscosity coefficient |
| $Z$ | Blade number |
| $\beta_2$ | Blade outlet angle |
| $\delta$ | Blade thickness |
| $\rho$ | Density |
| $\eta_h$ | Flow efficiency |

| $\eta_v$ | Volume efficiency |
|---|---|
| $\Delta_{pd}$ | The friction loss of impeller disk |
| $\Delta$ | Radient operator |
| $\varphi$ | Blade wrap angle |
| $\varphi_0$ | Tongue angle |
| $\omega_N$ | Precision of torque meter |
| $\omega_p$ | Precision of pressure meters |
| $\omega_{\eta_h}$ | Uncertainty of hydraulic efficiency |
| $\omega_{\dot{Q}}$ | Precision of turbine meter |

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
