# Peer review of "Performance Optimization of High Specific Speed Centrifugal Pump Based on Orthogonal Experiment Design Method"

_processes, doi:10.3390/pr7100728_

Round 1

Reviewer 1 Report

  This paper aims to model the influence of impeller parameters on the  efficiency of centrifugal pumps using orthogonal design experiments. The topic of this paper is of interest to the community of mechanical engineering and hydraulics. The experiments are conducted in a compact and rigorous manner. The authors covered in a skilful way the pertinent relevant literature. The results are fully sustained by the numerical experiments and illustrated in the manuscript.

I recommend a very minor revision of the manuscript, before its publication.

The boundary conditions and the initial conditions of the model must be inserted after Eq. (2). An English language spell check is required.

Reviewer 2 Report

The article has almost perfect structure. I have written “almost”, because the paper is too long and should be made shorter. In my opinion it is not necessary to provide commonly known equations (Navier-Stockes, turbulence models, etc), that can be found in appropriate references. Generally the methodology of the experimental and numerical researches are described without doubts.

Another my remark deal with picture description -  many pictures consist of many subpictures with indicators (a, b, c,…) described in text. I think – for better understanding – the mining of the indicators should be provided in picture description as well.

I have some doubts about choosing impeller parameters for an experimental scheme. It is commonly known how is an influence of the impeller outlet parameters on the efficiency and cavitation performance. In other words – the authors took into account factors that have the least impact on the impeller operation. In my opinion more interested are impeller inlet parameters (width b1, attack angle, an angle of overlap, etc). Nevertheless it is authors decision and the paper is worth of publication.
